# Epoxy Composites Reinforced with ZnO from Waste Alkaline Batteries

**DOI:** 10.3390/ma15082842

**Published:** 2022-04-13

**Authors:** Isaac Lorero, Mónica Campo, Carmen Arribas, Margarita Gonzalez Prolongo, Felix Antonio López, Silvia G. Prolongo

**Affiliations:** 1Materials Science and Engineering Area, ESCET, Rey Juan Carlos University, C/Tulipán s/n, 28933 Madrid, Spain; isaac.lorero@urjc.es (I.L.); monica.campo@urjc.es (M.C.); 2Materials and Aerospacial Production, Politechnic University, Plaza del Cardenal Cisneros, 3, 28040 Madrid, Spain; carmen.arribas@upm.es (C.A.); mg.prolongo@upm.es (M.G.P.); 3National Centre for Metallurgical Research (CENIM), Spanish National Research Council (CSIC), Avda. Gregorio del Amo, 8, 28040 Madrid, Spain; f.lopez@csic.es

**Keywords:** recycle of battery waste, epoxy composite, ceramic oxide particles

## Abstract

The zinc alkaline battery is one of the most popular sources of portable electrical energy, with more than 300,000 tons being consumed per year. Accordingly, it is critical to recycle its components. In this work, we propose the use of zinc oxide (ZnO) microparticles recovered from worn-out batteries as fillers of epoxy resins. These nanocomposites can be used as protective coatings or pigments and as structural composites with high thermal stability. The addition of ceramic nanofillers, such as ZnO or/and TiO_2_, could enhance the thermal and mechanical properties, and the hardness and hydrophobicity, of the epoxy resins, depending on several factors. Accordingly, different nanocomposites reinforced with recycled ZnO and commercial ZnO and TiO_2_ nanoparticles have been manufactured with different nanofiller contents. In addition to the different ceramic oxides, the morphology and size of fillers are different. Recycled ZnO are“desert roses” such as microparticles, commercial ZnO are rectangular parallelepipeds nanoparticles, and commercial TiO_2_ are smaller spherical nanoparticles. The addition of ceramic fillers produces a small increase of the glass transition temperature (<2%), together with an enhancement of the barrier effect of the epoxy resin, reducing the water diffusion coefficient (<21%), although the maximum water uptake remains constant. The nanocomposite water absorption is fully reversible by subsequent thermal treatment, recovering its initial thermomechanical behavior. The water angle contact (WCA) also increases (~12%) with the presence of ceramic particles, although the highest hydrophobicity (35%) is obtained when the epoxy resin reinforced with recycled flowerlike ZnO microparticles is etched with acid stearic and acetic acid, inducing the corrosion of the ZnO on the surface and therefore the increment of the surface roughness. The presence of desert rose ZnO particles enhances the de lotus effect.

## 1. Introduction

The recycling of batteries is an important challenge in present-day society, and it was identified by the European Commission on the Strategic Action Plan on Batteries: Building a Strategic Battery Value in Europe, in April of 2019 [1]. The estimated European market could be worth up to EUR 250 billion annually from 2025 onwards, due to the higher demand for electric vehicles, among others. In addition to researching more effective recycling processes of spent batteries, the use of their recycled products must be encouraged for the production of batteries and other new applications. An alkaline battery includes many components, and, therefore, its recycling allows for the recovery of several valuable ingredients, such as zinc oxide (ZnO) [2].

ZnO is an interesting material with many applications, such as electronics, solar cells, photocatalysis, and sensors. However, the synthesis of ZnO particles is expensive, due to the closure of many mines and deficits in raw and polluting processes to synthesize it [3]. Hence, the synthesis of ZnO microparticles from waste provides an interesting sustainable solution.

In this work, the ZnO particles are incorporated into epoxy thermosetting resins in order to obtain high thermal and mechanical performance. The incorporation of fillers involves several inherent problems, mainly challenges associated with their dispersion. Recently, the possibility of manufacturing the nanoparticles in situ [4,5,6] has been researched, and the results have shown better dispersion. Moreover, a good interphase with the matrix is achieved using a one-stage process. However, our main challenge is the search for new, useful applications of recycled ZnO particles. Accordingly, an easy dispersion technique, free of solvent and with low energy consumption, is used to manufacture the ceramic composites. 

The incorporation of ZnO fillers into a thermosetting matrix can achieve nanocomposites with enhanced thermal, electrical and mechanical behavior [7,8,9,10,11]. These oxide nanoparticles have an attractive array of electrical, thermal, optical and mechanical properties. Their main applications are as coatings with high corrosion and UV resistance, enhanced abrasion, and wear properties. It is worthy to note that ZnO particles are non-toxic materials, producing environmentally friendly coatings. Moreover, in particular, ZnO nanoparticles are widely used as fillers due to their excellent ability to absorb ultraviolet radiations. In addition to coatings, ZnO nanoparticles are also incorporated into an epoxy matrix of glass-fiber-reinforced epoxy nanocomposites [11] to enhance their toughness. The great affinity between this ceramic nanofiller and the epoxy matrix increases the matrix-dominated properties, such as impact strength. 

It has been reported [8,12] that the curing process of a thermosetting resin could be affected by the presence of ceramic microscaled and nanoscaled particles, confirming a negative effect, which induces a decrease in the crosslinking degree of the resin. However, M. Ghaffari et al. [13] have demonstrated the contrary effect: ZnO addition leads to a decrease in the activation energy of the curing reaction because the reduction becomes more pronounced as the particles size shifts from the micro- to the nanoscale. This influence is associated with the catalytic effect of ZnO, forming a complex between Zn^2+^ and oxirane rings. 

On the other hand, ZnO/epoxy coatings can breed hydrophobic surfaces [14,15,16,17]. Rough structures and low surface energy contribute to hydrophobicity [18]. Rough and low energy surfaces are usually built with hierarchical micro and nanostructures. In particular, raspberry-like particles are quite suitable for manufacturing hydrophobic surfaces [19]. To benefit from the advantages of hierarchical ceramic structures, chemical etching of the polymer matrix is required. One of the most common treatments is based on stearic acid or/and acetic acid [17,20,21]. In fact, the hydrophobicity could be controlled by modifying the stearic concentration, which influences the surface roughness of the epoxy thermoset [17,20]. 

Umapathy et al. [10] confirmed that the mechanical properties of ZnO/epoxy nanocomposites depend on the filler content. In fact, the tensile, flexural and impact strength increases with the ZnO content up to a maximum. Then, a negative trend appears due to the agglomeration problems. 

Considering all these presented advantages, ceramic particle/epoxy composites are being applied in civil and defense constructions as coatings on the surface of the neat epoxy thermosets and composites to reduce their water absorption and improve their surface mechanical behavior [22,23,24,25]. The matrix aging, low toughness, and low energy impact strength of polymer composites usually hinder further development due to the occurrence of microcracks on the surface. For example [23], the swaying motion of composite transmission line tower caused by wind might lead to surface micro-cracks, which accelerate the hydrothermal aging of a polymer matrix composite, significantly reducing the lifetime of the structure, which would be replaced in advance. The addition of ceramic nanoparticles within the surface of the epoxy thermoset improves the ability to resist crack initiation and propagation [22]. Moreover, this coating could reduce the wettability of the surface and water uptake, thus avoiding this problem and increasing the lifetime of the composite structures.

The geometry and size of fillers play an important role in the obtained properties of the composite. A recent study [26] based on an epoxy matrix filled with alumina has been published, in which the influence of the size (micro- and nano-scale) and morphology (rods and spherical nanoparticles) of the ceramic nanofillers on the mechanical properties is analyzed. To summarize, the ceramics rods improve the stiffness and toughness while the tensile properties are better for spherical nanoparticles, but this trend strongly depends on the amount of the ceramic load added. 

Finally, in addition to the size and morphology of the ceramic fillers, the nature of the oxide also affects the performance of the nanocomposite. Oxides of aluminum, titanium and zinc, among others, are usually incorporated into epoxy thermosets [27]. Structural coatings are used for enhancing their mechanical and thermal properties. The nature of the ceramic fillers seems not to have a strong influence, while the size and the morphology of the particles is more influential, as well as the content and the dispersion degree. 

In this study, an aeronautical epoxy thermosetting resin is filled with different ceramic micro- and nano-particles, based on ZnO and TiO_2_. One of the ZnO particles comes from the recycling of exhausted alkaline batteries. These nanoparticles have been previously characterized by Field Emission Gun of Scanning Electron Microscopy (FEG-SEM) and Infrared Fourier Transform Spectroscopy (FTIR) [28]. The main goal of this work is to find possible advantages to, and to study the performance of, recycled ZnO with regard to other expensive commercial ceramic nanoparticles as fillers of an epoxy thermosetting resin, commonly used as a matrix of composites manufactured by resin infusion techniques for aeronautical structures or as protective coatings. Moreover, this study allows for the evaluation of the influence of the size, morphology and nature of the ceramic oxide filler in the mechanical, thermal and ageing properties of the composites.

## 2. Materials and Methods

### 2.1. Materials and Sample Preparation

The selected epoxy resin, whose commercial name is Araldite LY 556, is based on diglycidyl ether of bisphenol A (DGEBA), and it is cured by an aromatic hardener, XB 3473, in a stoichiometric ratio (100:23 *w*:*w*). Both were supplied by Huntsman (Huntsman International LLC., Houston, TX, USA). The curing treatment consists of heating the homogenous mixture at 140 °C for 8 h. Previously, it was confirmed that this treatment is suitable to reach the maximum epoxy conversion. 

The studied samples were manufactured in bulk by pouring in mold. The thickness of the mold was 2 mm. Then, they were cut in different specimens as a function of the geometry and size required by each test. 

Different ceramic particles are used as fillers. Microparticles recovered from the black mass of exhausted alkaline batteries by the National Centre for Metallurgical Research (CENIM-CSIC) [29] are used, whose characterization has been already published [30]. They present a wurtzite structure and high purity. As a reference, two commercial ceramic oxide nanoparticles, ZnO and TiO_2_, have been selected. Both were supplied by Sigma Aldrich, with an average diameter lower than 100 nm.

Ceramic composites were manufactured by molding. Previously, the ceramic particles were sonicated into the epoxy prepolymer at 50 Hz and 50% of amplitude for 1 h. Afterward, the dispersion was degassed 15 min at 80 °C. The percentage of nanoparticles added was 6 and 10 wt%. These contents were selected according to the bibliography [10,26] and by considering the geometry of the filler. Nanocomposites with rods or nanotubes usually require lower concentration [26]. The neat epoxy resin is transparent, but it becomes opaque via the addition of the fillers.

To increase the hydrophobicity of the nanocomposites’ surfaces, they were treated by a published chemical etching [20,31]. The sanded samples were immersed subsequently in two media: first, in 2 M acetic acid in ethanol, and then by immersion in 5 wt% stearic acid for 1 h per solution. The treated surfaces were then dried at room temperature for 12 h.

### 2.2. Characterization

The morphology and size of the ceramic oxides, ZnO and TiO_2_, were determined by Field Emission Gun Electron Scanning Microscopy (FEG-SEM, Nova Nano SEM230 model, FEI, Hillsboro, OR, USA). 

The curing behavior of ZnO/epoxy composites was studied by the measurement of the glass transition temperature (T_g_) measured by Differential Scanning Calorimetry (DSC (mod.822e, Mettler Toledo, Greifensee, Switzerland). The heating rate was 10 °C/min from 20 to 220 °C in a nitrogen atmosphere. Three consecutive scans were made to evaluate the possible postcuring. The first scans of the cured samples do not show any exothermal peaks, indicating that they were fully cured.

To analyze the effect of hydrothermal aging, the samples were immersed in distilled water at 40.0 ± 0.1 °C to induce accelerated water diffusion. Before immersion tests, the samples were dried at 50 °C for one week. The dimensions of the specimens used were 35 × 12.4 × 1.4 mm, which allowed them to be later analyzed by by Dynamic Mechanical Thermal Analysis (DMTA). 

Two specimens of each sample were immersed in distilled water, and they were periodically removed from water, carefully wiped, and weighted with an accuracy of ±0.01 mg. The period for which the sample was out of the water was less than 1 min. The water sorption (water uptake) at the time, *t*, was calculated as:(1)$$M_{t}\left(\%\right)=\left(\frac{W_{t}-W_{0}}{W_{0}}\right)100$$
being *W*_0_ the initial weight of the dry specimen and *W_t_* the weight of the hydrothermally aged specimen at each time, *t*.

Dynamic mechanical thermal analysis (DMTA) was used to characterize unaged samples and to investigate the effects of hydrothermal aging on their thermomechanical properties. Experiments were performed in dual cantilever bending mode using a DMTA V Rheometric Scientific instrument. Measurements were done at 1 Hz, and the temperature was increased from 30 °C to 220 °C at 2 °C/min. The specimen dimensions were 35 × 12.4 × 1.4 mm. The elastic or storage modulus (*E*′), the loss modulus (*E*″) and the loss tangent (tan δ) were recorded as a function of temperature. The maximum in tan δ vs. temperature curves was determined to identify the α-relaxation associated with the glass transition. 

Finally, the hydrophobicity of the non-treated and of the etched surfaces was determined by the measurement of the contact angle of a water drop, using a goniometer (Tamé-Har 200 mod. p/n 200-F1, Syuccasunna, NJ, USA). The surface profile was measured by an optical perfilometer (Zeta 20 model from KLA-Tencor, Milpitass, CA, USA). 

## 3. Results

Figure 1 shows FEG-SEM images of the studied ceramic particles. Commercial nanoparticles have a smooth surface. Commercial TiO_2_ nanoparticles (Figure 1a) are spherical, with an average diameter lower than 100 nm, and form large agglomerates. They present both, anatase and rutile phases. Commercial ZnO nanoparticles (Figure 1b) are rectangular parallelepipeds, whose sides of the square base do not exceed 100 nm, while their height reaches up to 500 nm. On the contrary, the geometry of the recycled ZnO particles (Figure 1c) are similar to the desert roses with a diameter of around 2.5 µm. Each petal is a nanosheet with a very homogeneous thickness of about 35 nm. In both, ZnO fillers present a wurtzite structure.

ZnO nanoparticles can present different morphologies, such as nanowires, nanorods, nanobelts, desert roses and spherical nanoparticles. In the bibliography, it is not clear which is the best morphology of ZnO particles to be used as a polymer filler [26,27]. The hierarchical structure observed for the recycled ZnO particles could promote the increase of their photocatalytic activity and their hydrophobic effect, which are enhanced by their nanosized subunits [31]. In spite of their micro-scale size, the specific area of the ceramic particles is significantly increased by their hierarchical structure, which also raises the interphase region with the epoxy matrix. Accordingly, the high hydrophobicity of polymer composites is ensured by adding micro- or nanoparticles with hierarchical morphologies, such as the recycled ZnO particles, to find the “Lotus effect” [19]. 

Manufactured epoxy composites were characterized by DSC and DMTA. Figure 2 shows the glass transition temperature (T_g_) of epoxy thermosets reinforced with different contents of ceramic particles measured by three consecutive DSC scans. The initial T_g_ of neat epoxy resin is 139 °C and lightly increases for the next scans up to 142 °C due to the higher efficiency of scanning thermal curing. This means that the curing treatment applied is suitable. The addition of ceramic particles induces a T_g_ decrease, which is more pronounced for commercial nanoparticles. This is explained by the hindering of the curing reaction [9,32], which is more significant for smaller nanoparticles, as is the case with the commercial ones. The steric hindrance of the nanoparticles limits the reaction of curing, so the thermoset forms a less crosslinked epoxy network, lowering the T_g_. The obtained results confirmed that the steric hindrance is more effective when the added filler is a nanoparticle than microparticle. This is because the nanoparticles can be located in the free volume of the freshly generated thermosetting network, hindering the advance of the curing process. This fact can be also explained by the increase of the viscosity of the mixture by the addition of nanoparticles, which limits the diffusion of reactants. The same reason also explains the higher decrement of T_g_ as a function of ceramic content. 

Finally, the consecutive thermal DSC treatments trigger a recovery of T_g_ in all cases, due to the scanning temperature being higher than T_g_, enhancing the postcuring treatment. In any case, the final T_g_ of epoxy composites is lower than the neat epoxy thermoset, and this decrease is more marked with the increase of ceramic particles content added. The T_g_ reduction of composites is explained by several reasons [9]. One of them is that the nanofillers size is larger than the free volume of the thermosetting network, producing a slide between the chains, which results in increased free volume. In contrast, the T_g_ decrease is also related to the enhanced polymer chain dynamics associated with extra free volume at the matrix–filler interface.

One of the possible advantages to introduce ceramic particles into the epoxy thermoset is to enhance its hydrothermal strength. Figure 3 shows the water absorption curves (M_t_ (%) versus t^½^·h^−1^, being h the specimen thickness) obtained for the samples reinforced with both ZnO particles, recycled and commercial ones. All the samples were immersed in distiller water for one year. Two specimens were tested by each studied sample, and the absorption curves were almost coincident. 

In all studied systems, the addition of ceramic oxide particles induces a very slight decrease of the maximum absorbed water, named *M*_∞_ (Table 1). This fact is associated with the physical barrier behavior of added fillers. The lower water uptake values were obtained for the commercial nanoparticles, whose lower size implies a higher specific surface, enhancing its barrier properties.

Water absorption appears to follow a two-stage process. An initial stage (fast diffusion) is followed by a slow gradual increase in weight gain. This behavior can be described by a two-stage diffusion model [33,34]. The first stage follows Fickian diffusion, and the second stage is assumed to be controlled by relaxation, that is, the water absorbed enhanced the polymer structural relaxation, thus the overall water uptake is given by:(2)$$M_{t}=M_{\infty }\left(1+k\sqrt{t}\right)\;\left\{1-exp\left[-7.3{\left(\frac{Dt}{h^{2}}\right)}^{0.75}\right]\right\}$$
where *M*_∞_ and *D* are the quasi-equilibrium moisture uptake level and diffusion coefficient associated with the first stage of diffusion, respectively; *k* is a constant related to the polymer relaxation due to water uptake; and *h* is the specimen thickness.

The experimental data of all the samples follow the same pattern. The values of *M*_∞_, *D* and *k* are summarized in Table 1. As can be seen, increasing the nanoparticles content leads to the decrease of *D*. The presence of ceramic particles forces water to follow a tortuous pathway lowering the diffusion coefficient. This means that in spite of the amount of absorbed water, the water absorption rate is decreased on the epoxy composites. The fine influence of ceramic particles on the hydrothermal aging of nanocomposites is associated with a weak polymer-particles interface [35,36] because the diffusion of water into the epoxy composites depends on the polymer matrix structure, water–polymer interactions and the possible degradation processes resulting from them [37].

The *k* values are roughly constant (1.8 × 10^−4^ ± 0.1 × 10^−4^ s^−1/2^). It seems that the presence of ceramic particles does not affect the structural relaxation of the epoxy network that takes place in the second stage.

The effect of water uptake on the thermomechanical behavior of the samples was determined by DMTA. In order to evaluate the reversibility of the water absorption phenomenon, after hydrothermal aging, the samples were dried in an oven at 45 °C for 12 days. The next figure shows the DMTA curves: the variation of storage modulus (*E*′) and loss tangent (tan δ) as a function of temperature for the neat epoxy thermoset (Figure 4a,b) and, as an example, for the composite reinforced with 6 wt% recycled ZnO (Figure 4c,d) in the three different stages: as manufactured, hydrothermally aged and dried after hydrothermal aging. As it is expected, both of the initial samples—the neat epoxy resin and the composites—present only one narrow tan δ peak, corresponding to the α-relaxation of the thermosetting network. Accordingly, the storage modulus, *E*′, remains approximately constant in the glassy state and suffers a marked drop due to the glass transition to the rubbery stage. DMTA curves obtained for aged samples present a different shape. The tan δ peak shows a clear shoulder at low temperature, which is associated with the plasticized resin by absorbed water. Nevertheless, the DMTA curves of dried samples recover their original shape, confirming that the water uptake is reversible. The absorbed water is removed by a thermal treatment, meaning that most of it is free water, occupying the free volume of the network. There is hardly any bonded water, which cannot be removed by evaporation. This is an interesting result because the aging problems of polymers are associated with the possible negative effect of the absorbed water, and, in the studied system, these consequences are not permanent. At this point, it is worth noting that the glass transition temperature is affected for the water uptake, while no significant differences are observed on the storage modulus in the glassy state (Figure 4d), meaning that the stiffness of the neat epoxy thermoset and composites remains practically unaltered by the hydrothermal aging. This means that the water ageing affects the thermal strength but not the mechanical behavior of the studied materials. 

Figure 5 collects the glass transition temperature, measured as the maximum of tanδ peak, for all studied samples. For the hydrothermally aged samples, two T_g_s are measured for the plasticized region (shoulder) and for the non-modified region (peak). In all cases, the tendency is the same. The T_g_ of the epoxy matrix is recovered after the drying treatment, indicating that the water absorption process is reversible [9]. However, the aged composites with absorbed water present two differentiated regions, non-modified and plasticized, with two glass transitions.

The recovery of the glass transition after drying implies that the absorbed water is free water, meaning that any permanent bond is formed between the water molecules and the chemical groups of the network, only weak hydrogen interactions linked the water with the polar groups of the epoxy resin, such as the hydroxyl groups formed by the oxirane ring opening. These polar interactions with the water justify the appearance of a new lower T_g_ in the aged nanocomposites. Comparing different studied composites, it seems that the highest decrease of the T_g_ associated with water plasticization is measured for the thermoset reinforced with TiO_2_. In fact, this is the system with the lowest T_g_ recovery after drying. This indicates that the water absorption and its corresponding plasticization effect depends more on the ceramic oxide nature than the size or the morphology of particles added. This could be explained by the weaker interphase formed for the nanocomposites filled with TiO_2_ in comparison with ZnO. 

Finally, the surface hydrophobicity of the different samples was determined by the measurement of the contact angle of water (WCA), whose results are shown in Figure 6. The original epoxy coating surfaces are smooth and possess hydrophilic wettability, with a WCA of 61°. The addition of ceramic particles induces an increment of hydrophobicity due to the low surface energy. In this case, the most efficient filler is the recycled ZnO. Moreover, in addition to low energy, the hydrophobicity also requires the presence of surface roughness. Accordingly, the epoxy matrix is chemically etched, inducing an important increment of WCA, close to 85°, for the composite reinforced with recycled “desert rose” ZnO particles. Figure 7 shows the surface profile for some of the studied samples. The average roughness of non-treated epoxy thermoset is 1.30 ± 0.16 µm. The roughness lightly increases with the ceramic particle addition in the range of 1.4–1.6 µm. However, when the chemical etching treatment is applied on the surface of the composites, their roughness significantly increases in the range of 3.2–4.2 µm. The roughest surfaces correspond to the composites reinforced with ZnO and etched with acetic acid and stearic acid. On them, the maximum WCA is measured. This effect has been previously observed by other authors [20], and it is explained by the corrosion of ZnO particles on the surface by acetic acid, resulting in an increase of the surface roughness. Wu et al. [38] have recently confirmed that, under acidic conditions, the non-polar long chain alkyl group of stearic acid reacted with the hydroxyl group in acetic acid. At this point, the metal ions of the ZnO were displaced to the stearic acid and generated globular zinc stearate between [Zn^2+^] and [CH_3_COO^−^], justifying the increase of the roughness and the hydrophobic properties. This effect is more pronounced for recycled particles due to their micro-scale size. Therefore, the higher hydrophobicity for the etched epoxy composites reinforced with recycled ZnO particles is due to the most efficient chemical treatment in the presence of ZnO filler, increasing the roughness of the surface, as well as the morphology of desert rose of recycled ZnO, enhancing the lotus effect. 

## 4. Conclusions

New reuse is explored for ZnO particles obtained by the recycling of spent alkaline batteries; they are added as fillers in epoxy coatings. To analyze their performance, the obtained results are compared with two other commercial ceramic oxide nanoparticles, TiO_2_ and ZnO. The main differences between the studied ZnO particles are their geometry and size. Recycled ZnO are bigger desert roses, in contrast with the regular and non-hierarchical structures in the commercial ZnO nanoparticles.

The addition of these ceramic fillers scarcely affects the thermal strength and mechanical stiffness of the thermoset, measured by the T_g_ and storage modulus. Moreover, the studied epoxy composites show the same hydrothermal resistance as the neat epoxy thermoset. The presence of ceramic oxide particles lightly reduces the water diffusion coefficient. However, the contact angle of water drop, which allows one to study the surface hydrophobicity, is increased by the oxide ceramic particles. This increment is more pronounced with the chemical treatment of the surface with stearic and acetic acid. The composite reinforced with recycled ZnO particles shows the highest WCA and, therefore, hydrophobicity due to its highest roughness, which is caused by the oxidation with acids and the lotus effect, associated with its desert rose morphology. 

## Figures and Tables

**Figure 1 materials-15-02842-f001:**
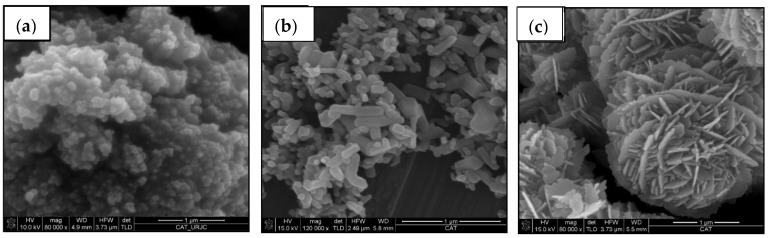
FEG-SEM micrographs of ceramic particles: (**a**) commercial TiO_2_, (**b**) commercial ZnO and (**c**) recycled ZnO.

**Figure 2 materials-15-02842-f002:**
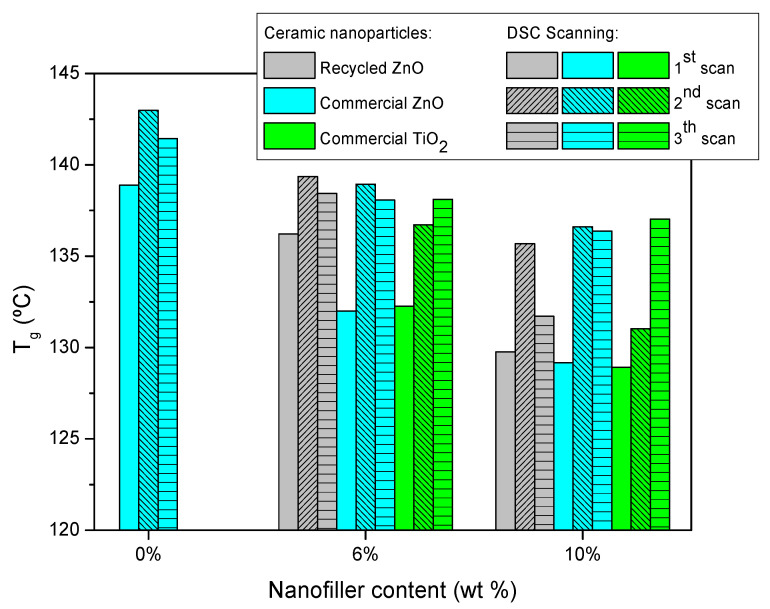
Glass transition temperature, measured by three consecutive DSC scans, for the studied epoxy composites reinforced with ceramic nanoparticles.

**Figure 3 materials-15-02842-f003:**
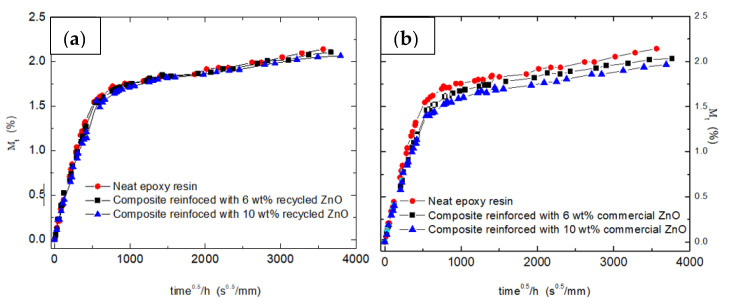
Absorption curves for neat epoxy thermoset and epoxy nanocomposites reinforced with recycled (**a**) and commercial (**b**) ZnO particles.

**Figure 4 materials-15-02842-f004:**
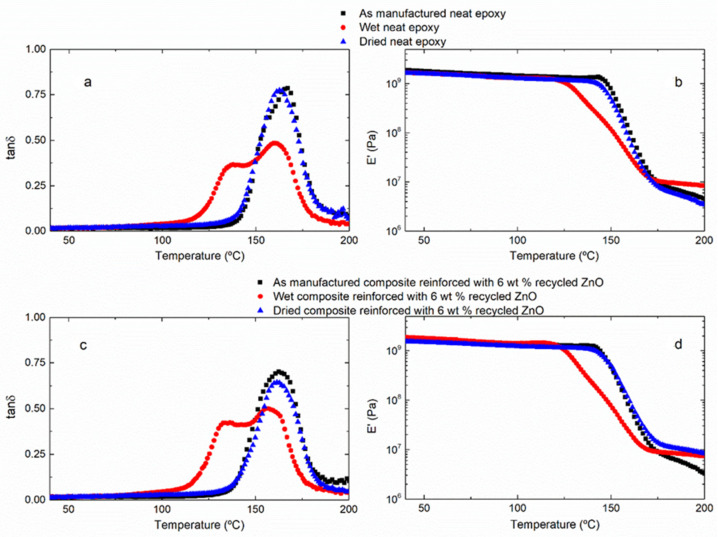
DMTA curves of neat epoxy thermoset (**a**,**b**) and composites reinforced with 6 wt% recycled ZnO (**c**,**d**) in different three stages: initial, aged and dried.

**Figure 5 materials-15-02842-f005:**
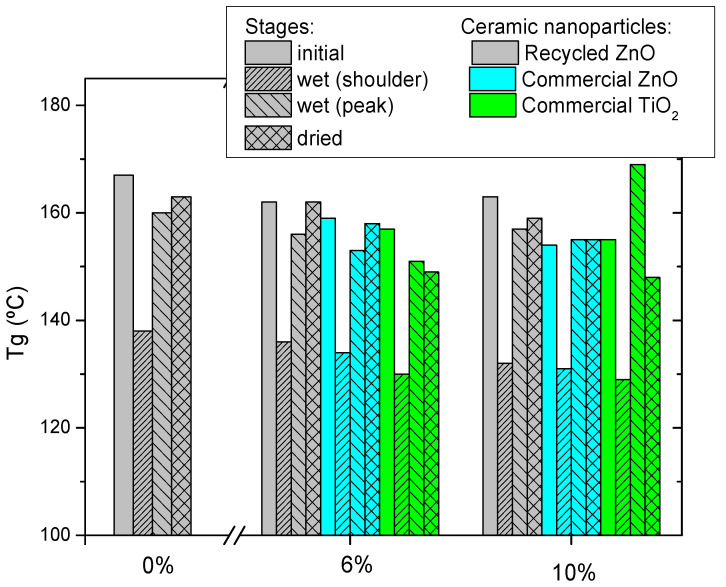
T_g_ determined as the maximum of the tan δ peak for the studied composites reinforced with ceramic particles in different stages: as manufactured (initial), after hydrothermal aging (wet) and after the thermal drying treatment (dried).

**Figure 6 materials-15-02842-f006:**
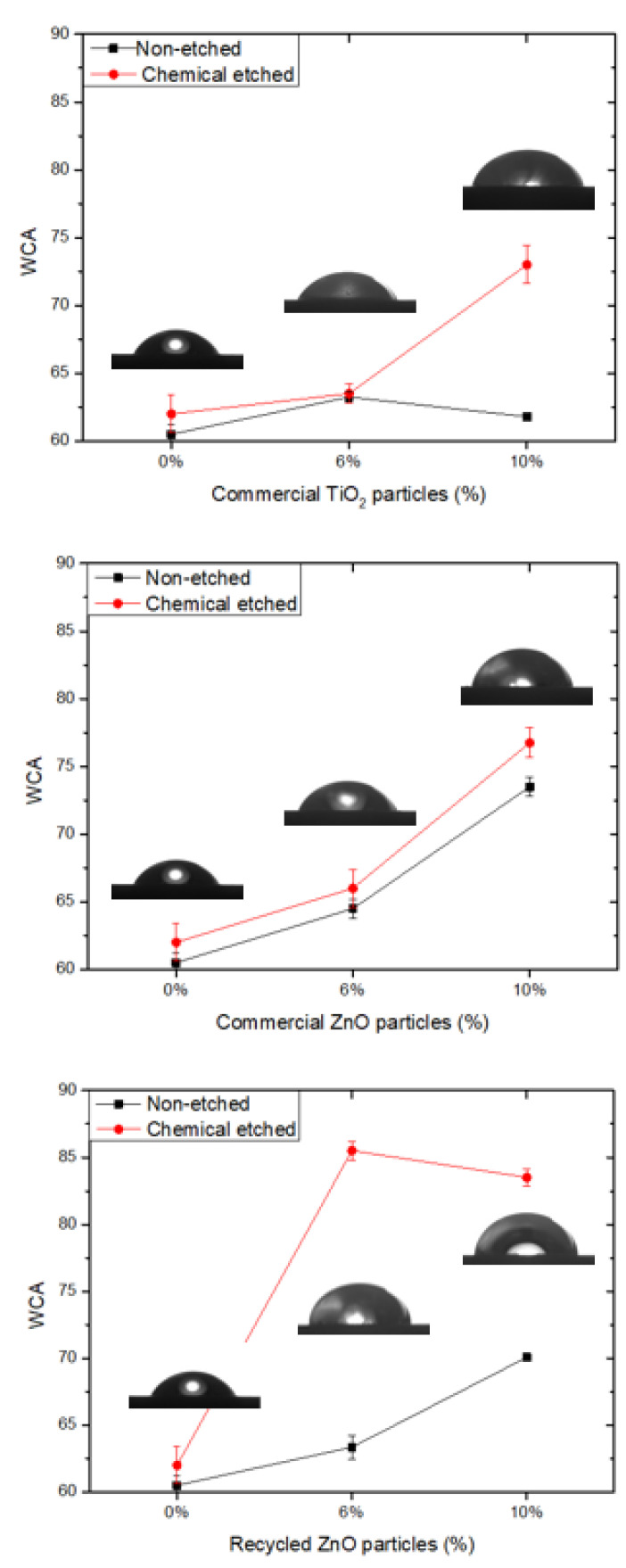
Static water contact angle (WCA) of chemically etched and non-treated composites and epoxy thermoset surfaces. The drop images correspond to etched surfaces.

**Figure 7 materials-15-02842-f007:**
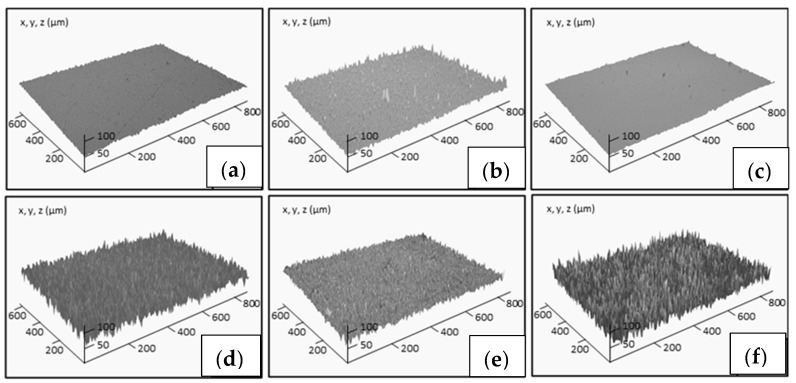
Surface profiles of non-treated surfaces of neat epoxy thermoset (**a**) and composites reinforced with 10 wt% commercial TiO_2_ (**b**) and commercial ZnO (**c**), and etched surfaces of nanocomposites with 6 wt% commercial TiO_2_ (**d**), commercial ZnO (**e**) and recycled ZnO (**f**).

**Table 1 materials-15-02842-t001:** Water absorption parameters of epoxy and nanocomposites.

Nanoparticle	Nanofiller Content (wt%)	*M*_∞_ (wt%)	*D*. 10^7^ (mm^2^·s^−1^)	*k*. 10^4^ (s^−1/2^)
-	0	1.64 ± 0.03	7.5 ± 0.3	1.9 ± 0.1
Recycled ZnO	6	1.67 ± 0.03	6.8 ± 0.1	1.7 ± 0.1
	10	1.64 ±0.03	6.2 ± 0.2	1.7 ± 0.1
Commercial ZnO	6	1.57 ± 0.03	6.5 ± 0.2	1.8 ± 0.1
	10	1.52 ± 0.03	6.4 ± 0.2	1.8 ± 0.1
Commercial TiO_2_	6	1.63 ± 0.03	6.5 ± 0.1	1.8 ± 0.1
	10	1.60 ± 0.03	5.9 ± 0.2	1.7 ± 0.1

## Data Availability

Not applicable.

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
