# Peer review of "Epoxy Composites Reinforced with ZnO from Waste Alkaline Batteries"

_materials, 2022, doi:10.3390/ma15082842_

Round 1

Reviewer 1 Report

Review report on the topic ‘Epoxy composites reinforced with ZnO from waste alkaline batteries’. Comments are listed below:

  1. The abstract should be improved deeply because most of the abstract mentions the common type of information and and the present work is not focused.
  2. The novelty of the work should be discussed first in respect of the application.
  3. The introduction section is very weak. Add more references related to the current work and define the gap clearly. Also add few recent papers in the introduction section (published in 2021 and 2022).
  4. The experimental setup is missing. Add the images of the samples.
  5. The discussion section is very poor, and it looks like a technical report. Add more discussion in support of results along with proper references.
  6. Can authors review the theoretical modeling of the Epoxy Composite Filled with nanoparticles?
  7. Add the following recent articles in the reference list.

https://doi.org/10.1016/j.matpr.2018.06.526

https://doi.org/10.1007/s10965-021-02774-w

https://doi.org/10.1007/s10965-020-02359-z

Author Response

Comments to the Reviewer #1

  1. The abstract has been modified deeply, removing general information and including particular information about the present work.

Abstract: The zinc alkaline battery is one of the most popular source of portable electrical energy, being consumed more than 300,000 tons per year. For it, it is critical to recycle its components. In this work, we propose the use of zinc oxide (ZnO) microparticles recovered from worn out batteries as filler of epoxy resins. These nanocomposites can be used as protective coatings or pigments and structural composites with high thermal stability. The addition of ceramic nanofillers, such as ZnO or/and TiO2, could enhance the thermal and mechanical properties, the hardness and hydrophobicity of the epoxy resins, depending on several factors. For it, different nanocomposites reinforced with recycled ZnO and commercial ZnO and TiO2 nanoparticles have been manufactured with different nanofiller contents. In addition to the different ceramic oxides, the morphology and size of fillers is different. Recycled ZnO are“desert roses” like microparticles, commercial ZnO are rectangular parallelepipeds nanoparticles and commercial TiO2 are smaller spherical nanoparticles.

The addition of ceramic fillers produces a small increase of the glass transition temperature (< 2%) together with an enhancement of the barrier effect of the epoxy resin, reducing the water diffusion coefficient (< 21%) although the maximum water uptake remains constant. The nanocomposite water absorption is fully reversible by subsequent thermal treatment, recovering their initial thermomechanical behaviour. The water angle contact (WCA) also increases (~12 %) with the presence of ceramic particles, although the highest hydrophobicity (35 %) is obtained when the epoxy resin reinforced with recycled flowerlike ZnO microparticles is etched with acid stearic and acetic acid, inducing the corrosion of the ZnO on the surface and therefore the increment of the surface roughness. The presence of desert rose ZnO particles enhances de lotus effect.  

  1. Now, in the revised manuscript, the novelty of the work has been discussed, focused on the main application.

In this work, the ZnO particles are incorporated into epoxy thermosetting resins in order to get high thermal and mechanical performance. The incorporation of fillers involves several inherent problems mainly challenges associated to their dispersion. Recently, it is being researching about the possibility to manufacturing the nanoparticles in situ [4-6], obtaining interesting results because a better dispersion. Moreover, a good interphase with the matrix is achieved using an one stage process. However, our main challenge is the search of new useful applications of recycled ZnO particles. For it, an easy dispersion technique, free of solvent and with low energy consumption, is used to manufacture the ceramic composites.

In this study, an aeronautical epoxy thermosetting resin is filled with different ceramic micro- and nano-particles, based on ZnO and TiO2. One of the ZnO particles comes from the recycling of exhausted alkaline batteries. These nanoparticles have been previously characterized by Field Emission Gun of Scanning Electron Microscopy (FEG-SEM) and Infrared Fourier Transform Spectroscopy (FTIR) [28]. The main goal of this work is to find possible advantages and performance of recycled ZnO with regard to other expensive commercial ceramic nanoparticles as filler of an epoxy thermosetting resin, commonly used as matrix of composites manufactured by resin infusion techniques for aeronautical structures or as protective coatings. Moreover, this study allows evaluating the influence of the size, morphology and nature of the ceramic oxide filler in the mechanical, thermal and ageing properties of the composites.   

  1. The Introduction has been modified and extended. Recent references have been included, as recommended by several reviewers.

The recycling of batteries is an important challenge in present-day society, as it was identified by the European Commission on the Strategic Action Plan on Batteries: Building a Strategic Battery Value in Europe, in April of 2019 [1]. The estimated European market could be worth up to EUR 250 billion annually from 2025 onwards, due to the higher demand for electric vehicles, among others. In addition to researching more effective recycling processes of spent batteries, the use of their recycled products must be encouraged for the production of batteries and other new applications. An alkaline battery includes many components and therefore, its recycling allows for the recovery of several valuable ingredients, such as Zinc oxide (ZnO) [2]. 

ZnO is an interesting material with many applications, such as electronics, solar cells, photocatalysis, sensors… However, the synthesis of ZnO particles is expensive, due to the closure of many mines and deficit in raw, and polluting processes to synthesize it [3]. Hence, the synthesis of ZnO microparticles from waste provides an interesting sustainable solution.

In this work, the ZnO particles are incorporated into epoxy thermosetting resins in order to get high thermal and mechanical performance. The incorporation of fillers involves several inherent problems mainly challenges associated to their dispersion. Recently, it is being researching about the possibility to manufacturing the nanoparticles in situ [4-6], obtaining interesting results because a better dispersion. Moreover, a good interphase with the matrix is achieved using an one stage process. However, our main challenge is the search of new useful applications of recycled ZnO particles. For it, an easy dispersion technique, free of solvent and with low energy consumption, is used to manufacture the ceramic composites.           

The incorporation of ZnO fillers into a thermosetting matrix can achieve nanocomposites with enhanced thermal, electrical and mechanical behavior [7-11]. These oxide nanoparticles have an attractive array of electrical, thermal, optical and mechanical properties. Their main applications are as coatings with high corrosion and UV resistance together with enhanced abrasion, and wear properties. It is worthy to note that ZnO particles are non-toxic materials, producing environmental friendly coatings. Moreover, in particular, ZnO nanoparticles are widely used as fillers due to their excellent ability to absorb ultraviolet radiations. In addition to coatings, ZnO nanoparticles are also incorporated into epoxy matrix of glass fiber reinforced epoxy nanocomposites [11] to enhance their toughness. The great affinity between this ceramic nanofiller and epoxy matrix increases the matrix-dominated properties, such as impact strength.     

It has been reported [8,12] that the curing process of a thermosetting resin could be affected by the presence of ceramic microscaled and nanoscaled particles, confirming a negative effect, which induces a decrease in the crosslinking degree of the resin. However, M. Ghaffari et al. [13] have demonstrated the contrary effect: ZnO addition leads to a decrease in the activation energy of the curing reaction, being the reduction more pronounced as the particles size range from the micro- to the nanoscale. This influence is associated with the catalytic effect of ZnO, forming a complex between Zn2+ and oxirane rings.

On the other hand, ZnO/epoxy coatings can breed hydrophobic surfaces [14-17]. Rough structures and low surface energy contribute to hydrophobicity [18]. Rough and low energy surfaces are usually built with hierarchical micro and nanostructures. In particular, raspberry like particles are quite suitable for manufacturing hydrophobic surfaces [19]. To benefit from the advantages of hierarchical ceramic structures, chemical etching of polymer matrix is required. One of the most common treatments is based on stearic acid or/and acetic acid [17,20, 21]. In fact, the hydrophobicity could be controlled by modifying the stearic concentration, which influences the surface roughness of the epoxy thermoset [17,20].

Umapathy et at. [10] confirmed that the mechanical properties of ZnO/epoxy nanocomposites depend on the filler content. In fact, the tensile, flexural and impact strength increases with the ZnO content up to a maximum. Then, a negative trend appears due to the agglomeration problems. 

Considering all these presented advantages, ceramic particle/epoxy composites are being applied in civil and defense constructions as coatings on the surface of the neat epoxy thermosets and composites to reduce their water absorption and improve their surface mechanical behavior [22-25]. The matrix aging, low toughness, and low energy impact strength of polymer composites usually hinder further development due to the occurrence of microcracks on the surface. For example [23], the swaying motion of composite transmission line tower caused by wind might lead to surface micro-cracks, which accelerates the hydrothermal aging of a polymer matrix composite, significantly reducing the lifetime of the structure, which would be replaced in advance. The addition of ceramic nanoparticles within the surface of the epoxy thermoset improves the ability to resist crack initiation and propagation [22]. Moreover, this coating could reduce the wettability of the surface and water uptake, avoiding this problem and increasing the lifetime of the composite structures.

The geometry and size of fillers play an important role on the obtained properties of the composite. A recent study [26] based on an epoxy matrix filled with alumina has been published, in which the influence of the size (micro- and nano-scale) and morphology (rods and spherical nanoparticles) of the ceramic nanofillers on the mechanical properties is analyzed. Summarizing, the ceramics rods improve the stiffness and toughness while the tensile properties are better for spherical nanoparticles, but this trend strongly depends on the amount of the ceramic load added.

Finally, in addition to the size and morphology of the ceramic fillers, the nature of the oxide also affects to the performance of the nanocomposite. Oxides of aluminum, titanium and zinc, among others, are usually incorporated into epoxy thermosets [27]. Structural coatings for enhancing their mechanical and thermal properties. The nature of the ceramic fillers seems not to have a strong influence, being more decisive the size and the morphology of the particles, as well as the content and the dispersion degree.                

In this study, an aeronautical epoxy thermosetting resin is filled with different ceramic micro- and nano-particles, based on ZnO and TiO2. One of the ZnO particles comes from the recycling of exhausted alkaline batteries. These nanoparticles have been previously characterized by Field Emission Gun of Scanning Electron Microscopy (FEG-SEM) and Infrared Fourier Transform Spectroscopy (FTIR) [28]. The main goal of this work is to find possible advantages and performance of recycled ZnO with regard to other expensive commercial ceramic nanoparticles as filler of an epoxy thermosetting resin, commonly used as matrix of composites manufactured by resin infusion techniques for aeronautical structures or as protective coatings. Moreover, this study allows evaluating the influence of the size, morphology and nature of the ceramic oxide filler in the mechanical, thermal and ageing properties of the composites.   

  1. The experimental part has been improved. The images of samples have not been included in order to reduce the length of the paper. However, a clear description has been included.

Materials and sample preparation

The selected epoxy resin, whose commercial name is Araldite LY 556, is based on diglycidyl ether of bisphenol A (DGEBA) and it is cured by an aromatic hardener, XB 3473, in a stoichiometric ratio (100:23 w:w). Both were supplied by Huntsman. The curing treatment consists of heating the homogenous mixture at 140ºC for 8 hours. Previously, it was confirmed that this treatment is suitable to reach the maximum epoxy conversion. 

The studied samples were manufactured in bulk by pouring in mold. The thickness of the mold was 2 mm. Then, they were cut in different specimens as a function of the geometry and size required by each test. 

Different ceramic particles are used as fillers. Microparticles recovered from the black mass of exhausted alkaline batteries by the National Centre for Metallurgical Research (CENIM-CSIC) [29] are used, whose characterization has been already published [30]. They present a wurtzite structure and high purity. As a reference, two commercial ceramic oxide nanoparticles, ZnO and TiO2, have been selected. Both were supplied by Sigma Aldrich, with an average diameter lower than 100 nm.

Ceramic composites were manufactured by molding. Previously, the ceramic particles were sonicated into the epoxy prepolymer at 50 Hz and 50% of amplitude for 1 hour. Afterward, the dispersion was degassed 15 min at 80ºC. The percentage of nanoparticles added was 6 and 10 wt %. These contents were selected according with the bibliography [10,26] considering the geometry of the filler. Nanocomposites with rods or nanotubes usually require lower concentration [26]. The neat epoxy resin is transparent but it becomes opaque by the addition of the fillers.

To increase the hydrophobicity of the nanocomposites surfaces, they were treated by a published chemical etching [20,31]. The sanded samples were immersed subsequently in two media: first, in 2 M acetic acid in ethanol and followed by immersion in 5 wt% stearic acid for 1 h per solution. The treated surfaces were then dried at room temperature for 12 h.

  1. The results have been discussed deeply, including references of other published works.

Figure 1 shows FEG-SEM images of the studied ceramic particles. Commercial nanoparticles have a smooth surface. Commercial TiO2 nanoparticles (Fig 1a) are spherical with an average diameter lower than 100 nm, which form large agglomerates. They present both, anatase and rutile phases. Commercial ZnO nanoparticles (Fig 1b) are rectangular parallelepipeds, whose sides of the square base do not exceed 100 nm while their height reaches up to 500 nm. On the contrary, the geometry of the recycled ZnO particles (Fig 1c) reminds of the desert roses with a diameter of around 2.5 µm. Each petal is a nanosheet with a very homogeneous thickness of about 35 nm. In both, ZnO fillers present wurtzite structure.

ZnO nanoparticles can present different morphologies, such as nanowires, nanorods, nanobelts, desert roses and spherical nanoparticles. In the bibliography, it is not clear which is the best morphology of ZnO particles to be used as a polymer filler [26,27]. The hierarchical structure observed for the recycled ZnO particles could promote the increasing of their photocatalytic activity and their hydrophobic effect, which are enhanced by their nanosized subunits [31]. In spite of their micro-scale size, the specific area of the ceramic particles is significantly increased by hierarchical structure, which also raises the interphase region with the epoxy matrix. For it, high hydrophobicity of polymer composites is ensured by adding micro- or nanoparticles with hierarchical morphologies, such as the recycled ZnO particles, looking for the “Lotus effect” [19].

Manufactured epoxy composites were characterized by DSC and DMTA. Figure 2 shows the glass transition temperature (Tg) of epoxy thermosets reinforced with different contents of ceramic particles measured by three consecutive DSC scans. The initial Tg of neat epoxy resin is 139 °C and lightly increases for the next scans up to 142 °C due to the higher efficiency of scanning thermal curing. This means that the curing treatment applied is suitable. The addition of ceramic particles induces a Tg decrease, which is more pronounced for commercial nanoparticles. This is explained by the hindering of the curing reaction [9,32], which is more significant for smaller nanoparticles, as it is the case of the commercial ones. The steric hindrance of the nanoparticles limits the reaction of curing, so the thermoset forms a less crosslinked epoxy network, lowering the Tg. The obtained results confirmed that the steric hindrance is more effective when the added filler is a nanoparticle than microparticle. This is because the nanoparticles can be located in the free volume of the freshly generated thermosetting network, hindering the advance of the curing process. This fact can be also explained by the increase of the viscosity of the mixture by the addition of nanoparticles, which limits the diffusion of reactants. The same reason also explains the higher decrement of Tg as a function of ceramic content.

The recovery of the glass transition after drying implies that the absorbed water is free water, meaning that any permanent bond is formed between the water molecules and the chemical groups of the network, only weak hydrogen interactions linked the water with the polar groups of the epoxy resin, such as the hydroxyl groups formed by the oxirane ring opening. These polar interactions with the water justify the appearance of a new lower Tg in the aged nanocomposites. Comparing different studied composites, it seems that the highest decrease of the Tg associated to water plasticization is measured for the thermoset reinforced with TiO2. In fact, this is the system with lowest Tg recovery after drying. This indicates that the water absorption and its corresponding plasticization effect depends more on the ceramic oxide nature than the size or the morphology of particles added. This could be explained by the weaker interphase formed for the nanocomposites filled with TiO2 in comparison with ZnO.

  1. The authors have revised the theoretical models of epoxy resins filled with nanoparticles and even the experimental works, recommend by other reviewers, in order to explain and understand the obtained experimental results.

  1. The references recommended by the reviewer are very interesting and some of them are included in the revised manuscript. In fact, one of them has been useful to describe the state of the art about the studied materials.

The geometry and size of fillers play an important role on the obtained properties of the composite. A recent study [26] based on an epoxy matrix filled with alumina has been published, in which the influence of the size (micro- and nano-scale) and morphology (rods and spherical nanoparticles) of the ceramic nanofillers on the mechanical properties is analyzed. Summarizing, the ceramics rods improve the stiffness and toughness while the tensile properties are better for spherical nanoparticles, but this trend strongly depends on the amount of the ceramic load added.

Ref 26. Verma, V., Sayyed, A. H. M., Sharma, C., Shukla, D. K. Tensile and fracture properties of epoxy alumina composite: role of particle size and morphology. Journal of Polymer Research 2020, 27, 388  

Reviewer 2 Report

The work is related to the development of epoxy composites. The main idea is to check the possibility of using the zinc oxide filler recovered from worn-out zinc alkaline batteries in the epoxy matrix. Obtained results and conclusions are well supported by experimental data and important for both the production of composite materials and zinc alkaline battery recycling technologies. However, some issues should be addressed before the recommendation of the acceptance of the manuscript to publication:

1) Some other related works (e.g. Thipperudrappa, S. et al. (2020). Influence of zinc oxide nanoparticles on the mechanical and thermal responses of glass fiber‐reinforced epoxy nanocomposites. Polymer Composites, 41(1), 174-181; Boopalan, M., et al. "A Study on the Mechanical Properties of Zinc Oxide Reinforced Epoxy Composites." Asian Journal of Chemistry 25.5 (2013): 2931) can be considered in the Introduction section.

2) In the experimental part, the authors made a reference to other works, which describe the characterization of the used fillers, however, I suggest that it will be better to include some major characteristics of used fillers and/or their comparison directly into the manuscript.

3) Please check the text of the manuscript for typos, e.g. lines 141-142, line 188 ("distilled water" is suggested), line 220, etc.

4) The phrase "Two samples were tested by each studied sample" (line 188) is difficult for understanding.

5) The quality of Fig. 1, Fig. 6, and Fig. 7 is low and it is impossible to read important data in them.

6) What is the error in WCA measurement? It is should be pointed out.

Author Response

Comments to the Reviewer #2

  1. Now, the Introduction has been extended with the recommend references, enriching the state of the art about epoxy nanocomposites reinforced with different ceramic nanofillers.

Ref 10. Boopalan, M., Michael, R.J.V, Yoganand, K. S., Umapathy, M.J. A Study on the Mechanical Properties of Zinc Oxide Reinforced Epoxy Composites. Asian Journal of Chemistry 2013, 25, 5, 2931-2932

Ref 11. Thipperudrappa, S., Kini, A. U., Hiremath, A. Influence of zinc oxide nanoparticles on the mechanical and thermal responses of glass fiber-reinforced epoxy nanocomposites. Polymer Composites 2020, 41, 174-181

Ref 26. Verma, V., Sayyed, A. H. M., Sharma, C., Shukla, D. K. Tensile and fracture properties of epoxy alumina composite: role of particle size and morphology. Journal of Polymer Research 2020, 27, 388

  1. In the revised manuscript, the characteristics and properties of the used nanofillers have been enlarged to facilitate the reading of this paper.

Figure 1 shows FEG-SEM images of the studied ceramic particles. Commercial nanoparticles have a smooth surface. Commercial TiO2 nanoparticles (Fig 1a) are spherical with an average diameter lower than 100 nm, which form large agglomerates. They present both, anatase and rutile phases. Commercial ZnO nanoparticles (Fig 1b) are rectangular parallelepipeds, whose sides of the square base do not exceed 100 nm while their height reaches up to 500 nm. On the contrary, the geometry of the recycled ZnO particles (Fig 1c) reminds of the desert roses with a diameter of around 2.5 µm. Each petal is a nanosheet with a very homogeneous thickness of about 35 nm. In both, ZnO fillers present wurtzite structure.

  1. The authors have checked the entire document and have revised all the typing mistakes.

To analyze the effect of hydrothermal aging, the samples were immersed in distilled water at 40.0 ± 0.1 °C to induce accelerated water diffusion. Before immersion tests, the samples were dried at 50 °C for one week. The dimensions of the specimens used were 35 x 12.4 x 1.4 mm3, which allowed them to be later analyzed by by Dynamic Mechanical Thermal Analysis (DMTA).

Two specimens of each sample were immersed in distilled water and they were periodically removed from water, carefully wiped, and weighted with an accuracy of ± 0.01 mg.

  1. This phrase has been modified for a better understanding.

Two specimens were tested by each studied sample.

  1. Figure 5 has been formatted

  1. Figures 1, 6 and 7 has been formatted and enlarged.

  1. The error of experimental WCA measurement has been included in the Figure 6.

Reviewer 3 Report

There is new interesting research work in the title of"Epoxy composites reinforced with ZnO from waste alkaline 2 batteries" which is well written by Lorero et.al

Here is few comments:

  1. in Figure 5 formatting is required
  2. In the section of introduction add some nanomatrial-epoxy based referances for eg (Advanced Materials 30 (33), 1801523; Advanced Materials 31 (51), 1901802; ACS Applied Materials & Interfaces 12 (24), 27555-27561)
  3. In Figure 2 formatting is required
  4. In Figure 2 formatting is required

Author Response

Comments to the Reviewer #3

  1. Figure 5 has been formatted

  1. As recommend by the review, new references about epoxy nanomaterials are included in the Introduction section.

Ref 4. Pansare, A.V., Chhatre S.Y., Khairkar, S.R., Bell, J.G., Barbezat M., Chakarbarti, S. Nagarkar, A.A. Applied Materials & Inter-faces 2020, 12, 27555-27561

Ref 5. Pansare, A.V., Khairkar, S.R., Shedge, A.A., Chhatre S.Y., Patil, V.R., Nagarkar, A.A. In situ nanoparticle embedding for authentication of epoxy composites. Advanced Materials 2018, 30, 1801523

  1. Figure 2 has been formatted.

Round 2

Reviewer 1 Report

Accept in the present form.